# Autumn Olive (*Elaeagnus umbellata* Thunb.) Berries Improve Lipid Metabolism and Delay Aging in Middle-Aged *Caenorhabditis elegans*

**DOI:** 10.3390/ijms25063418

**Published:** 2024-03-18

**Authors:** Yebin Kim, Seonghyeon Nam, Jongbin Lim, Miran Jang

**Affiliations:** 1Department of Institute of Digital Anti-Aging Healthcare, Inje University, Gimhae 50834, Republic of Korea; dpqls1812@oasis.inje.ac.kr; 2Department of Food Bioengineering, Jeju National University, Jeju 63243, Republic of Korea; seonghyeon_n@stu.jejunu.ac.kr (S.N.); jongbinlim@jejunu.ac.kr (J.L.); 3Department of Food Technology and Nutrition, Inje University, Gimhae 50834, Republic of Korea

**Keywords:** autumn olive (*Elaeagnus umbellata* Thunb.), middle-aged *C. elegans*, lipid metabolism, delaying aging

## Abstract

This study evaluated the positive effects of autumn olive berries (AOBs) extract on delaying aging by improving lipid metabolism in middle-aged *Caenorhabditis elegans* that had become obese due to a high-glucose (GLU) diet. The total phenolic content and DPPH radical scavenging abilities of freeze-dried AOBs (FAOBs) or spray-dried AOBs (SAOBs) were examined, and FAOBs exhibited better antioxidant activity. HPLC analysis confirmed that catechin is the main phenolic compound of AOBs; its content was 5.95 times higher in FAOBs than in SAOBs. Therefore, FAOBs were used in subsequent in vivo experiments. FAOBs inhibited lipid accumulation in both the young adult and middle-aged groups in a concentration-dependent manner under both normal and 2% GLU conditions. Additionally, FAOBs inhibited ROS accumulation in a concentration-dependent manner under normal and 2% GLU conditions in the middle-aged worms. In particular, FAOB also increased body bending and egg production in middle-aged worms. To confirm the intervention of genetic factors related to lipid metabolism from the effects of FAOB, body lipid accumulation was confirmed using worms deficient in the daf-16, atgl-1, aak-1, and akt-1 genes. Regarding the effect of FAOB on reducing lipid accumulation, the impact was nullified in daf-16-deficient worms under the 2% GLU condition, and nullified in both the daf-16- and atgl-1-deficient worms under fasting conditions. In conclusion, FAOB mediated daf-16 and atgl-1 to regulate lipogenesis and lipolysis in middle-aged worms. Our findings suggest that FAOB improves lipid metabolism in metabolically impaired middle-aged worms, contributing to its age-delaying effect.

## 1. Introduction

The prevalence of obesity is increasing at an alarming rate worldwide, especially in recent decades [1]. The rise in obesity is also associated with an increased incidence of significant health risk factors and diseases, including insulin resistance, type 2 diabetes, non-alcoholic fatty liver disease, atherosclerosis, and certain cancers [2]. Obesity is influenced by the processes of lipogenesis and lipolysis [3,4]. *Caenorhabditis elegans* is considered an excellent multicellular organism model for studying the mechanisms of aging because it has a short lifespan, rapid progeny reproduction, and well-defined genetic pathways [5]. Furthermore, molecular pathways involved in energy metabolism in mammals, including humans, are strongly conserved in *C. elegans* [6]. Dietary glucose influences physiological and molecular processes in *C. elegans*, making it a valuable model for understanding human hyperglycemia and obesity. It has been reported that high-glucose diets (HGD) result in lipid accumulation, alter membrane fluidity, affect lifespan and progeny, and increase cellular levels of reactive oxygen species (ROS) and protein glycosylation [7,8]. 

Indeed, middle-aged people who have maintained unhealthy eating habits are susceptible to a variety of negative health outcomes [9]. Obesity in middle age is more sensitive to arteriosclerosis than obesity in youth [10]. In particular, in the case of middle-aged women who have reached menopause, abdominal and hepatic fat accumulation as well as waist circumference increase due to changes in hormonal and metabolic profiles. Middle-aged women with metabolic disturbances have an increased risk of cardiovascular disease and acute coronary artery disease due to excessive production of very low-density lipoprotein (VLDL), and even have a poorer prognosis than men [11]. However, although there are many reports of short-term diet effects with test periods set to days or weeks in in vivo studies, few studies have experimentally investigated accumulating dietary effects over middle age.

Autumn olive (*Elaeagnus umbellata* Thunb.) is one of the wild spiny branched shrubs, and is a plant in the Elaeagnaceae family that is native to Asia [10,11]. Autumn olive berries (AOBs) mature between September and November to an edible dark red color, and are sweet, sour, and juicy [12,13,14]. AOBs are known to contain a lot of catechins, lutein, gallic acid, caffeic acid, phytoene, phytofluene, β-cryptoxanthin, β-carotene, and α-cryptoxanthin [15,16]. Phytochemicals in AOBs, and particularly catechins, have been reported to exhibit antidiabetic effects by regulating glucose levels [17,18,19]. On the other hand, there are few studies on their anti-obesity effects related to lipid metabolism. Multiple reports explained that catechins are the main compounds of AOBs; meanwhile, such catechins are well-known compounds that have anti-obesity effects [20]. Thus, we hypothesized that catechins of AOBs might show an inhibitory effect on lipid accumulation.

In this experiment, AOBs were prepared for use as the main ingredients for powdered tea, and the following were investigated. First, we assessed the effect of AOBs on lipid accumulation in nematodes fed a high-glucose diet by middle age. Second, we evaluated the effect of autumn olive berries on menopausal symptoms (reduced behavior, decreased fertility, ROS accumulation) in middle age. Finally, we studied whether the autumn olive berries effect was affected by the intervention of lipid metabolism (lipogenesis and lipolysis)-related genetic factors.

## 2. Results

### 2.1. Confirmation of the Antioxidant Activity and Polyphenol Contents of FAOBs and SAOBs 

The total phenolic contents (TPC) of freeze-dried AOBs (FAOBs) and spray-dried AOBs (SAOBs) were 258.23 ± 1.00 mg GAE/100 g and 63.02 ± 2.00 mg GAE/100 g, respectively (Table 1). The DPPH scavenging capacity of FAOB and SAOB were 3842.23 ± 294.91 mg AA/100 g and 3151.78 ± 395.08 mg AA/100 g, respectively (Table 1). 

### 2.2. Catechin Identification

The HPLC chromatograms of FAOB and SAOB are shown in Figure 1. The results show that catechin was the key compound of FAOB. Interestingly, the catechin peak appeared high in FAOB and low in SAOB. The SAOB results almost coincided with the chromatogram of methanol used as the mobile phase. Quantification of catechin in FAOB and SAOB showed that they contained 154.02 ± 5.29 and 27.53 ± 3.05 mg/100 g, respectively (Table 2). There were statistically significant differences in the catechin content of FAOB and SAOB (*p*-value = 0.002), suggesting that differences in the physiological activities of FAOB and SAOB might be attributed to the catechin content. Based on the previously indicated polyphenol and catechin contents and antioxidant capacity results, we considered that FAOB has more potential, and thus decided to use FAOB for further experiments.

### 2.3. Safety of FAOB in C. elegans 

To determine the safe concentrations of AOB, an acute toxicity test was conducted with different concentrations (31.25–1000 μg/mL) of AOB. As a result, various tested concentrations of FAOB did not statistically show any difference in survival rates. However, survival rates decreased by 2% and 4% at 500 and 1000 μg/mL FAOB, respectively. Thus, safe concentrations (62.5, 125, and 250 μg/mL) of FAOB were used in subsequent experiments (Figure 2A). Also, FAOB at 250 μg/mL did not affect the survival of nematodes compared to the control, even under oxidative and heat stress conditions (Figure 2B,C).

### 2.4. FAOB Inhibits Lipid Accumulation in C. elegans

We examined the effect of AOB on fat accumulation in young and middle-aged adult worms. AOB showed a concentration-dependent decrease in total fat accumulation compared to the control under normal and 2% GLU diet conditions in young adult worms (Figure 3B). 

The inhibitory effect of FAOB on total fat accumulation was also confirmed in middle-aged worms (Figure 3C). Also, the triglyceride (TG) content in the AOB-treated group was reduced compared to the control under normal and 2% GLU diet conditions in the middle-aged group (Figure 3D).

### 2.5. AOB Reduces Reactive Oxygen Species (ROS) Accumulation in Middle-Aged C. elegans 

ROS are metabolites that cause aging, and it has been reported in previous studies that a high-GLU diet causes ROS accumulation [21]. FAOB exhibited a reduction effect on ROS production in both young and middle-aged worms (Figure 4). 

### 2.6. FAOB Exhibits an Age-Delaying Effect in C. elegans

In aging studies using nematodes, behavioral changes and offspring production ability are identified as biomarkers [22]. In our results, FAOB increased locomotion ability and fertility that were reduced by aging (Figure 5).

### 2.7. FAOB Regulates Lipid Accumulation by Daf-16 under 2% GLU Dietary Conditions

To investigate whether AOB regulates lipid metabolism in middle-aged worms under GLU feeding conditions, the total fat contents were analyzed using N2, daf-16, aak-1, akt-1, and atgl-1 knockdown worms. Although body fat reduction was found in the FAOB-treated group in wild-type N2 worms, such an effect of FAOB was not seen in daf-16 worms (Figure 6). Other worm strains had statistically reduced body fat by FAOB treatment, similar to N2. This means that the inhibitory effect of FAOB on body fat accumulation from a GLU diet depends on daf-16. 

### 2.8. FAOB Is Involved in Lipolysis Signaling Pathway under Fasting Conditions 

To investigate whether FAOB is involved in lipolysis in middle-aged worms during fasting, the total fat contents were analyzed using N2, daf-16, aak-1, akt-1, and atgl-1 knockdown worms. Body fat reduction was found in the FAOB treatment group compared to the control in wild-type N2 worms under fasting conditions (Figure 7). This effect of FAOB has not been seen in daf-16 and atgl-1 worms (Figure 7). In other worm strains, the body fat was statistically reduced by FAOB treatment, similarly to N2. Daf-16 and atgl-1 are factors involved in lipolysis signaling, and our results indicate that FAOB depends on daf-16 and atgl-1 to intervene in lipolysis.

## 3. Discussion

AOB is a red fruit that is reported to contain high levels of bioactive compounds such as carotenoids, organic acids, cinnamic acids, benzoic acids, flavonols, anthocyanins, tannins, and catechins [15,16,17]. We noted phenolic compounds among the bioactive substances in AOBs. Phenolic substances reported to be present in AOBs, including catechins, caffeic acid, chlorogenic acid, gallic acid, quercetin, and lutein, were used as reference compounds for the HPLC analysis. As a result, catechin was detected as the single main substance of AOBs we used, and the catechin content of FAOB was 5.95-fold higher than that of SAOB (Figure 1, Table 2). It has been reported that the phenolic contents and antioxidant activity of AOBs depend on the drying temperature during oven drying [15]. AOB was reported to have different contents of sugars, organic acids, and proteins depending on maturity [23]. Also, the content of minerals and organic acids differed depending on the presence or absence of the berry peels of *Ellaegnus angustifolia* L. [24]. In addition to AOB-related research, many studies have demonstrated that plant materials differ in their internal and external quality factors and potency as a result of their cultivation characteristics and post-harvest management [22,25]. On the other hand, plant extracts used in many studies are extracted using alcoholic solvents to obtain phenolic substances [26]; however, we performed water extraction because we intended to utilize AOB as a tea product. Taken together, it is sufficiently predictable that different pretreatment variables (post-harvest quality control, extraction, drying, and processing methods) of AOBs induce changes in their functionality. Therefore, further research is needed on the relationship between the composition profile and bioactivity of AOBs under different pretreatment conditions.

Catechins, widely recognized as the main functional components of tea and wine, include catechin, epicatechin, epigallocatechin (EGC), epicatechin-3-gallate (ECG), epigallocatechin-3-gallate (EGCG), and gallocatechin gallate (GCG) [27]. Catechins exhibit anti-cancer, anti-obesity, anti-diabetic, anti-cardiovascular, anti-infection, liver-protective, and nerve-protective properties [28,29]. In particular, catechin has antioxidant activity that scavenges ROS and at the same time can positively regulate lipid metabolism [28,29,30]. Therefore, it is believed that AOBs containing catechin as the abundant ingredient have the potential to effectively prevent obesity and related chronic degenerative diseases.

Lipid metabolism disorders and aging show close associations [21]. In particular, increased lipotoxicity under specific conditions, such as aging, may contribute to a variety of age-related diseases, including cardiovascular disease, cancer, arthritis, type 2 diabetes, and Alzheimer’s disease [21,31,32,33,34,35]. Recent studies have described the mechanisms of changes in lipolysis during aging [36,37]. However, although they can explain system-level changes in lipid metabolism, how these changes influence aging and aging-related diseases has not been verified [31]. Our results showed that during obesity induction with a high-GLU diet, middle-aged and young worms accumulated 20% and 40% lipid, respectively (Figure 3). It has been verified that more lipids accumulate with age when given the same high-GLU diet over the same period. 

There is an opinion that obesity, especially in middle age, can be caused not only by changes in the metabolic system, but also by a decrease in physical activity [38]. In the present study, day 7 worms were observed to experience a decline of more than 50% in physical activity compared to day 1 worms (Figure 5A). This result is consistent with the view that there is a decline in behavior in middle age that may lead to obesity. However, further research is required, as we have not confirmed whether the metabolic system at the molecular level has changed. 

Lipid metabolism is regulated by lipid synthesis (lipogenesis) and lipid degradation (lipolysis and beta-oxidation) [4,5]. These processes are intertwined at the molecular level and require a logical understanding to gain insight. Crucial in lipid metabolism is the working of daf-16/FOXO and atgl-1/ATGL [39,40,41]. In particular, FOXO is a key transcription factor in lipid metabolism that controls the balance of lipogenesis and lipolysis by downregulating ATGL, and is involved in lipid accumulation in mammals [41]. However, there are only a few reports on the interaction between daf-16/FOXO and atgl-1/ATGL in *C. elegans*. 

We prepared worms under a high-GLU diet or fasting conditions to clearly define the role of daf-16 and atgl-1 concerning the FAOB effect. As a result of FAOB administration, it was confirmed that the effect seen in wild-type worms disappeared in daf-16 knockdown worms under both GLU feeding and fasting conditions (Figure 6 and Figure 7). This indicates that FAOB is dependent on daf-16 related to lipogenesis and lipolysis. In addition, the effect of FAOB seen in N2 was lost in atgl-1 knockdown worms under fasting conditions, indicating that FAOB cooperates with atgl-1 under certain lipolytic conditions such as fasting. Previous studies have shown that daf-16 and atgl-1 can be regulated by akt-1/AKT and aak-1/AMPK [42,43,44,45]. However, our results confirmed that the FAOB effect is dependent on daf-16 and atgl-1, but is not related to akt-1 and aak-1 (Figure 6 and Figure 7). In other words, this suggests that FAOB acts directly on daf-16 or atgl-1, rather than daf-16 and atgl-1 being involved by regulating the upstream factors akt-1 or aak-1. Although daf-16 is upstream of atgl-1, further gene expression analysis is needed to clarify their upper and lower relationships for the effects of FAOB. 

## 4. Materials and Methods

### 4.1. Preparation of AOB Extracts 

The AOB samples used in this study were provided by NutriAdvisor (Seong-Nam, Gyeonggi-do, Republic of Korea). AOBs were grown in Hapcheon, Gyeongsangsam-do, Korea, and mature fruits were purchased in fresh condition and immediately freeze-dried or spray-dried. FAOBs were dried in a vacuum at −20 °C for 72 h with a freeze dryer (Ilshin Lab. Co., Gyeonggi-do, Republic of Korea). SAOBs were produced using a spray dryer (Fisher Scientific, Vernon Hills, IL, USA). The spray drying conditions were set to an injection temperature of 150 °C, an emission temperature of 100 °C, and a rotation speed of 14,000 rpm. Since the AOB samples were prepared for use as the main material for the tea powder, they were extracted under reflux at 80 °C with 100% distilled water (1:10 *w*/*v*) without using an organic solvent. After that, the extracts were evaporated to remove the solvent, and then made into powder. AOB samples were dissolved in DMSO to prepare a 15 mg/mL stock solution, and were stored at −80 °C and then diluted fresh in distilled water to an appropriate concentration right before the experiment. 

### 4.2. Reagents

All of the reagents used in our studies were HPLC or molecular biology grade. Reagents, unless otherwise specified, were obtained from Sigma Chemical Co. (St. Louis, MO, USA).

### 4.3. HPLC Analysis

To profile the phenolic compounds in AOB, information on the equipment, column, and analysis methods required for HPLC operation are presented in Table 3. The catechin used was D-catechin (CAS no. 154-23-4), which was dissolved in distilled water immediately before analysis. The catechin contents of the AOB samples were calculated using a catechin (5–50 μg/mL) calibration curve.

### 4.4. Total Phenolic Contents (TPC) of AOB

The total phenolic contents (TPC) in the AOBs were determined using the Folin–Ciocalteu (FC) method. Simply, 700 μL of the sample (mixed with FC in a 1:1 ratio) was combined with 700 μL of sodium carbonate and left in the dark for 1 h. The absorbance reading was performed at 720 nm. The total phenolic contents of the sample were calculated using the equation derived from the gallic acid standard curve. The results were expressed as gallic acid equivalents (GAE) per one hundred grams.

### 4.5. DPPH Radical Scavenging Capacity

In a simple procedure, a sample of the same volume was mixed with a DPPH solution (0.2 mM) and left at room temperature for 30 min. Then, the reaction mixture was measured at 517 nm using a microplate reader (SYNERGY HTX, Biotek, Santa Clara, CA, USA). 

The DPPH radical-scavenging activity was calculated using the following formula:radical scavenging activity (%) = [1 − (sample O.D./blank O.D.)] × 100

Furthermore, we determined the value of the antioxidant capacity, which was calculated using an ascorbic acid (AA) calibration curve and expressed in micrograms of AA one hundred per gram of sample dry weight (DW).

### 4.6. Worm Study

#### 4.6.1. Worm Culture

To conduct the in vivo experiments, we use *C. elegans* strain N2 (wild-type) and its derivative mutant strains; daf-16 (tm5030), atgl-1 (tm12352), aak-1 (tm1944), and akt-1 (tm399) were obtained from the National BioResource Project (NBRP) of Japan. 

Maintenance of the worms was applied by modifying what was recorded in the WormBook [46] to properly conduct our experiments. All of these strains were maintained on nematode growth medium (NGM) plates that were spread with *E. coli* OP50 and maintained at a temperature of 20 °C throughout the entire duration of the experiment. Age synchronization of the nematodes was achieved by separating the eggs from gravid adults using a solution comprising 6% sodium hypochlorite (Yuhanclorox, Seoul, Republic of Korea) and 5 M NaOH.

#### 4.6.2. Acute Toxicity

Toxicity assessments were performed with modifications to those recorded by the Organization for Economic Co-operation and Development (OECD) (2016) [47]. Synchronized L4 was washed twice with M9 buffer and suspended in M9 buffer containing cholesterol. Subsequently, 1 mL of this suspension was transferred to each well of a 24-well plate (20–30 worms per well) and combined with 10 μL of various concentrations of FAOB and SAOB. Then, the plates were incubated at 20 °C for 24 h. The acute toxicity results were quantified as percent survival after counting living worms.

#### 4.6.3. Determination of Stress Resistance

In the context of thermal and oxidative stress analysis, 50 synchronized N2 L1 larvae were prepared as follows: OP50 and FAOB were combined, and the mixture was seeded on NGM plates. The worm plate was first maintained at 20 °C for 60 h to assess the impact of thermal stress. Subsequently, it was incubated at 35 °C for 18 h. To investigate the effect on oxidative stress, the pretreated worms were transferred to a 24-well plate containing a 100 μM juglone (5-hydroxy-1,4-naphthoquinone) solution and incubated at 20 °C.

#### 4.6.4. Oil Red O Staining 

The oil red O assay was modified by Kim et al. [21]. L1 worms were exposed to different concentrations of extracts for 7 days, and cultivated worms were fixed in 4% formaldehyde for 24 h, then dehydrated with 60% isopropanol at −70 °C for 15 min. The dehydrated worms were washed three times with M9 buffer and dyed with an oil red O solution for 3 h. The stained worms were washed with M9 buffer, then observed under a microscope (Nikon, Seoul, Republic of Korea). The relative strength of the stained lipid droplets in the worms was quantified using Image J software (1.8.0, Joonas Regalis Rikkonen, Barcelona, Spain).

#### 4.6.5. Triglyceride (TG) Quantification Assay

The adult worms that were in culture were suspended in 500 μL of M9 buffer containing a 0.05% Tween-20 solution. Then, they were homogenized on ice for 5 min using a glass homogenizer (ALLSHENG, Hangzhou, China) to collect the pellet, which was subsequently centrifuged at 1000× *g* for 5 min. The obtained supernatant was analyzed for TG content. The TG content was measured via absorbance at 570 nm using a TG kit (Biomax, Gyeonggi-do, Republic of Korea).

#### 4.6.6. Reproduction and Pumping Rates

To count the number of progenies, the worms were transferred to fresh NGM plates daily throughout their reproductive period, and eggs were left on plates to hatch. The offspring of each worm were counted when they reached the L2 or L3 stage. The test was performed three times. 

To evaluate behavior activity, we counted the number of pharyngeal pumps on 0, 1, 5, and 7 days. Specifically, 10 nematodes were randomly selected for each concentration and age point, and the pumping frequency was determined three times for 20 s. This test was conducted three times.

#### 4.6.7. Determination of ROS Level

L1 worms were maintained in different concentrations of extracts for 7 days, and then incubated with 100 μM H_2_DCF-DA in the dark for 3 h. Subsequently, the nematodes were fixed on microscope slides using NAN3 (2%). These slides were observed using a Nikon ECLIPS Ci fluorescence microscope. The fluorescence intensities were examined using Image J software. An average of 10 worms per group was chosen for quantification.

### 4.7. Statistical Analysis

The results are presented as the means ± standard deviations (SD) of three independent replicates. The significance of intergroup differences was determined via one-way analysis of variance (ANOVA) followed by Tukey’s multiple range test. SPSS 27.0 was used for all statistical analyses except lifespan. *p*-values < 0.05 were considered to be significant. The survival results were analyzed with the Kaplan–Meier method, using the OASIS application (https://sbi.postech.ac.kr/oasis/, accessed on 29 November 2023). *p*-values of survival differences were determined with the log-rank test. Statistical analyses were performed using Student’s *t* test (unpaired, two tailed) with at least three replicates, unless otherwise indicated. Statistical analyses were performed in SPSS 27.0. 

## 5. Conclusions

This study investigated the relationship between obesity and aging in midlife, and examined the positive effects of AOB. The bio-activity of plants can vary depending on the drying method [24,48], and FAOB showed more effective radical scavenging ability compared to SAOB, which is believed to be because FAOB contained more polyphenols, including catechin. In the *C. elegans* study, treatment with FAOB significantly reduced lipid accumulation in both the young and the middle-aged groups. In addition, FAOB inhibited ROS accumulation in middle-aged worms under 2% GLU conditions. Interestingly, FAOB attenuated aging symptoms, including increasing the movement and reproductive capacity of aged worms. To understand the inhibitory effect of FAOB on lipid accumulation at the molecular level, we performed further investigations on the daf-16, aak-1, akt-1, and atgl-1 genes, which are related to lipid metabolism. The lipid accumulation reduction effect of FAOB was nullified under a high-GLU condition in the daf-16 knockdown mutant. The FAOB effect under fasting conditions was also nullified in the daf-16 and atgl-1 knockdown mutants. This suggests that these two factors are involved in the in vivo anti-obesity activity of FAOB in middle-aged worms, and further suggests its potential for anti-aging benefits. In summary, AOB acts on lipogenesis and lipolysis by regulating daf-16 and atgl-1 in middle-aged worms and alleviates aging symptoms. These results suggest that AOBs are a valuable material with the potential to delay aging by modulating lipid metabolism.

## Figures and Tables

**Figure 1 ijms-25-03418-f001:**
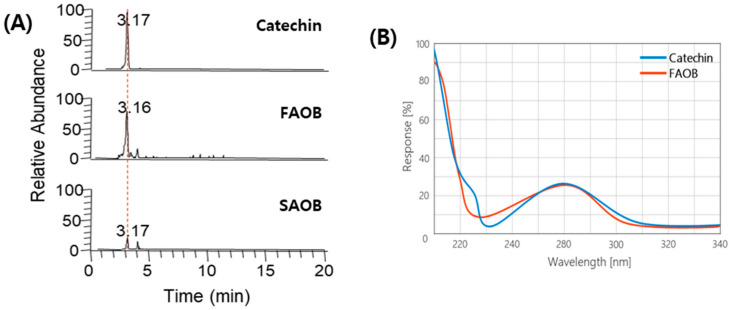
HPLC chromatograms of freeze-dried AOB (FAOB) and spray-dried AOB (SAOB). (**A**) HPLC chromatograms of catechin and two AOB extract samples. (**B**) Respective DAD spectra of the peaks with the same retention time. The red dotted line on the HPLC chromatogram represents the same retention time.

**Figure 2 ijms-25-03418-f002:**
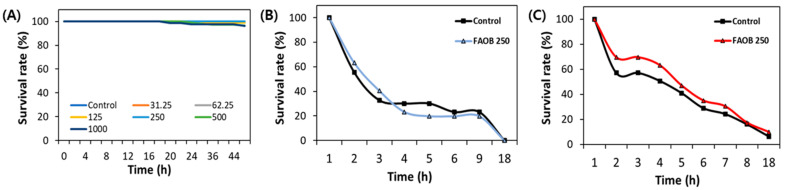
Safety of FAOB under various conditions. (**A**) The effect of AOB at different concentrations (31.25–1000 μg/mL) under normal conditions (50 worms were used in each group). The effect of FAOB at 250 μg/mL under (**B**) oxidative stress and (**C**) heat stress (*n* = 3 plates and 10 randomly selected worms were used).

**Figure 3 ijms-25-03418-f003:**
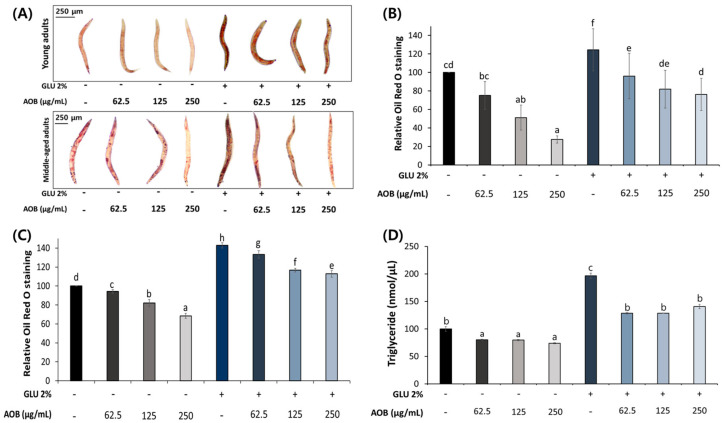
Inhibitory effect of AOB on lipid accumulation. (**A**) Images of stored body fat in young-aged and middle-aged worms under normal and 2% GLU conditions. Quantification results of total lipids in (**B**) young adult and (**C**) middle-aged worms under normal and 2% GLU dietary conditions. (**D**) Triglyceride (TG) contents of middle-aged worms under normal and 2% GLU conditions. Bars represent the means ± SDs of three separate experiments (for ORO assay, *n* = 3 plates and 10 randomly selected worms were used. More than 2000 worms were used in each group for TG assay). Different letters above the bars mean statistically significant differences (*p* < 0.05). “-” means untreated, and “+” means treated.

**Figure 4 ijms-25-03418-f004:**
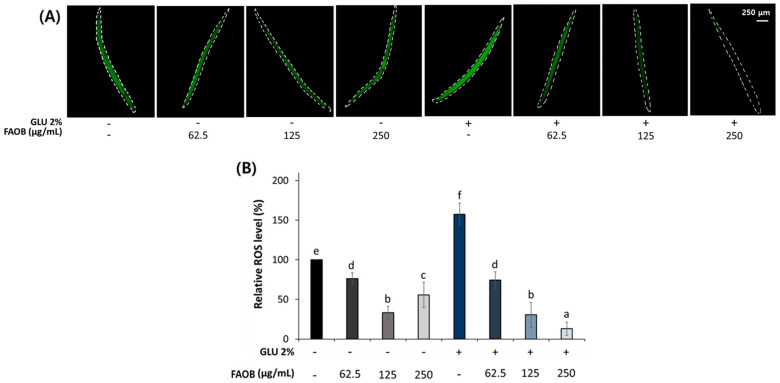
Inhibitory effect of FAOB on reactive oxygen species (ROS) accumulation. (**A**) Images of produced body ROS in middle-aged worms under normal and 2% GLU conditions. (**B**) Quantification results of ROS contents in middle-aged worms under normal and 2% GLU dietary conditions. Bars represent the means ± SDs of three separate experiments (*n* = 3 plates and 10 randomly selected worms were used). Different letters above the bars mean statistically significant differences (*p* < 0.05). “-” means untreated, and “+” means treated.

**Figure 5 ijms-25-03418-f005:**
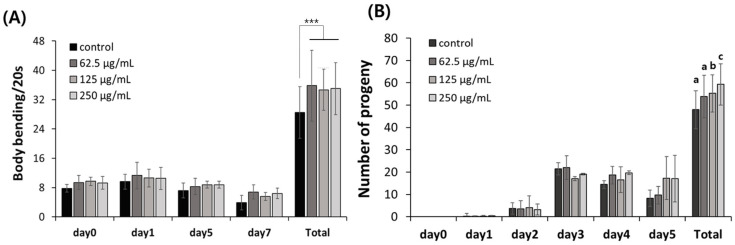
Age-delaying effects of AOB. (**A**) Bending and (**B**) reproduction abilities. Bars represent the means ± SDs of three separate experiments (*n* = 3 plates and 10 randomly selected worms were used). (**A**) Statistical analyses were performed using Student’s *t*-test (unpaired and two tailed). *** *p* < 0.001, (**B**) Statistical analysis was performed using one-way ANOVA followed by Tukey’s post hoc test. Different letters above the bars mean statistically significant differences (*p* < 0.05).

**Figure 6 ijms-25-03418-f006:**
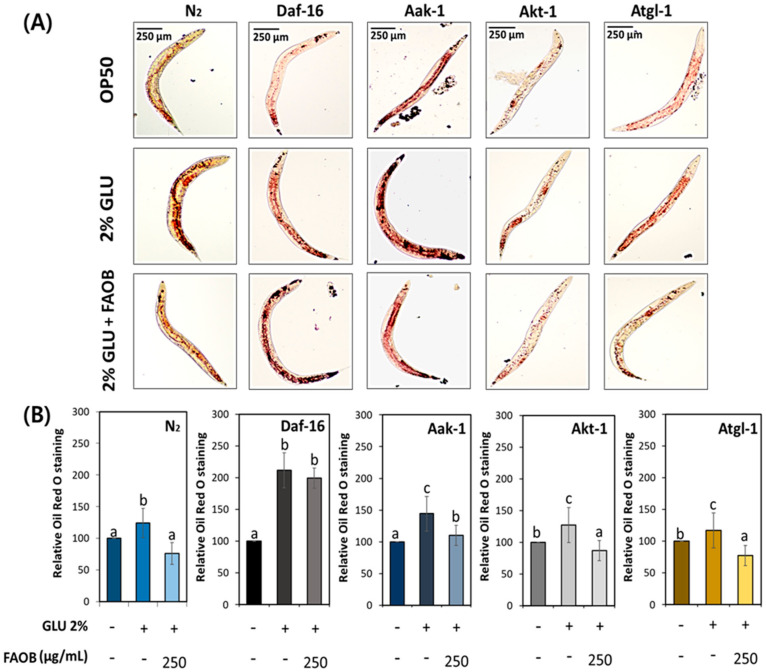
The effect of FAOB on lipid levels alters depending on lipid metabolism-related factors under 2% GLU dietary conditions. (**A**) Images of stored body fat in various worm strains under 2% GLU conditions. (**B**) Quantification results of total lipids in various worm strains under 2% GLU conditions. Bars represent the means ± SDs of three separate experiments (*n* = 3 plates and 10 randomly selected worms were used). Different letters above the bars mean statistically significant differences (*p* < 0.05). “-” means untreated, and “+” means treated.

**Figure 7 ijms-25-03418-f007:**
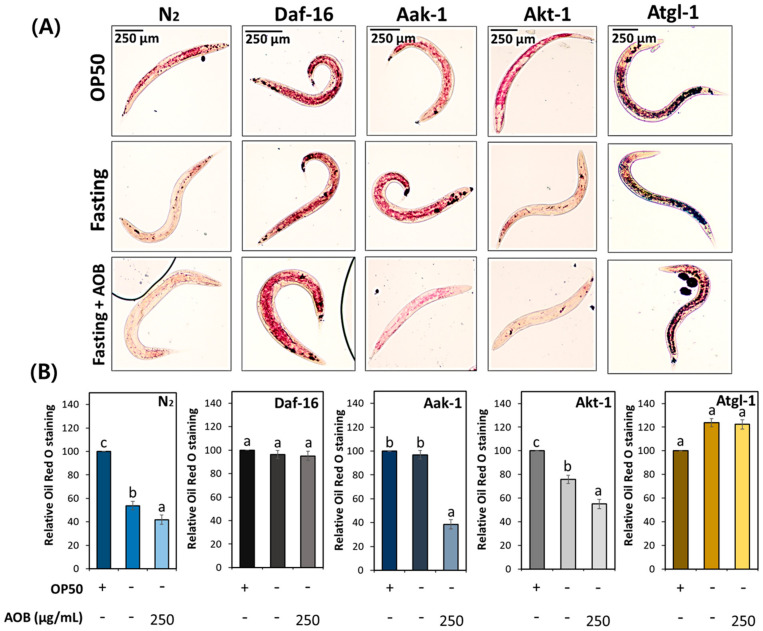
The effect of AOB on lipolysis alters depending on lipid metabolism-related factors under fasting conditions. (**A**) Images of stored body fat in various worm strains under fasting conditions. (**B**) Quantification results of total lipids in various worm strains under fasting conditions. Bars represent the means ± SDs of three separate experiments (*n* = 3 plates and 10 randomly selected worms were used). Different letters above the bars mean statistically significant differences (*p* < 0.05). “-” means untreated, and “+” means treated.

**Table 1 ijms-25-03418-t001:** TPC and DPPH values of freeze-dried AOBs (FAOBs) and spray-dried AOBs (SAOBs).

Samples	TPC (mg GAE/100 g)	DPPH (mg AA/100 g)
FAOB	258.23 ± 1.00	3842.23 ± 294.91
SAOB	63.02 ± 2.00	3151.78 ± 395.08
*p*-value *	0.05	0.03

TPC: total phenolic contents, GAE: gallic acid equivalent, and AA: ascorbic acid. All values represent the means ± SDs of three separate experiments. *: Statistical analyses were performed using Student’s *t*-test (unpaired and two-tailed).

**Table 2 ijms-25-03418-t002:** Catechin contents of freeze-dried AOB (FAOB) and spray-dried AOB (SAOB).

Samples	Catechin (mg/100 g)
FAOB	154.02 ± 5.29
SAOB	27.53 ± 3.05
*p*-value *	0.002

All values represent the means ± SDs of three separate experiments. *: Statistical analyses were performed using Student’s *t*-test (unpaired and two-tailed).

**Table 3 ijms-25-03418-t003:** HPLC analysis conditions.

HPLC (UltiMate 3000, Thermo Scientific, Sunnyvale, CA, USA)	
Column	Waters symmetry C18 column (4.6 × 150 mm, 5 µm)
Mobile phase	(A) 0.1% (*v*/*v*) aqueous phosphoric acid (B) Acetonitrile (Duksan, Ansan-si, Gyeonggi-do, Republic of Korea) Gradient method: 50–90% solvent B for 1 min, 90–85% solvent B for 2 min, 85–80% solvent B for 2 min, 80–70% solvent B for 3 min, 70–30% solvent B for 3 min, and a linear step from 30 to 10% solvent B for 9 min.
Flow rate	0.8 mL/min
Inject volume	20 μL
Detector	Diode array detector (DAD)
Wavelength	280 nm

## Data Availability

The data presented in this study are available on request from the corresponding author.

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
