# Peer review of "Autumn Olive (Elaeagnus umbellata Thunb.) Berries Improve Lipid Metabolism and Delay Aging in Middle-Aged Caenorhabditis elegans"

_ijms, 2024, doi:10.3390/ijms25063418_

Round 1

Reviewer 1 Report

Comments and Suggestions for Authors

            The paper investigates the potential positive effects of autumn olive berries (AOB) extract on aging delay by focusing on the improvement of lipid metabolism in middle-aged C. elegans, which have become obese due to a high glucose (GLU) diet. The study evaluates the antioxidant potential of freeze- and spray-dried AOB, with freeze-dried AOB demonstrating superior antioxidant capabilities, attributed to catechin as the main compound in the polyphenols of AOB. Subsequent in vivo experiments using freeze-dried AOB were conducted. The results overall imply that AOB may contribute to delaying aging by improving lipid metabolism in metabolic-impaired middle-aged worms.

            The paper covers an interesting topic with potential implications for understanding the role of natural compounds in lipid metabolism and aging. The manuscript is generally well-written, providing an overview of the research objectives, methods, and key findings. However, there are a few areas that need attention:

  1. Clarity of Results Presentation: The results need to be presented more clearly. Please check comments below.
  2. Figure Resolution: The figures accompanying the paper appear somewhat blurry. It is recommended to enhance the resolution of all figures to ensure that the data is presented clearly and is easily interpretable.
  3. English Language Corrections: The manuscript could benefit from a thorough proofreading to correct grammatical errors, improve sentence structure, and ensure overall language clarity. This is essential for enhancing the readability of the paper.
  4. Comments: Some comments also included that need to be considered:

1)    L3: “delays” should be corrected to “delay”

2)    L3: use the full name for “C. elegans” in the title, in the abstract and when you mention it for the first time in the main text

3)    L27: “contributes” should be corrected to “contribute”

4)    L54: “has effects” should be corrected to “have effects”

5)    L55: do you mean “to reduce obesity”?

6)    L63: use “phenolic” instead of “phenol”, as you did in Table 1

7)    L74: prefer “low” instead of “small”

8)    L86-87: omit the duplicate sentence

9)    L89-95: paragraph should be rewritten; 1) which figure shows that survival decreased in the concentration range of 500-1000 μg/mL of AOB? 2) you mention that safe concentrations used were 62.5, 125 and 250 (Figure 2A), but in Figure 2A all concentrations are shown and appear to be similar 3) since you used FAOB, you would better write FAOB 250 in Figures 2B and 2C 4) the survival of nematodes was affected, but not compared to the control, so be specific adding “compared to the control”

10) L102-108: 1) according to what you write under the Figure 3, Figure 3A, 3B and 3C inside the text should be changed to 3B, 3C and 3D, respectively 2) write TG in full name 3) TG Figure corresponds only to middle-aged worms according to what is written below, so be specific in lines 107-108

11) L110-117: 1) what “-” and ”+” mean? it isn’t mentioned anywhere 2) middle-aged worms are shown in Figure 3C? please correct it in line 112 3) correct “C” to “D” and add “under normal and 2% GLU condition” 4) write TG in full name for the first time

12) L119: write ROS in full name for the first time

13) L122: please improve the quality of images presented in Figure 4

14) L131: what do you mean “increased behavior”? please be more specific

15) L133: please improve the quality of images presented in Figure 5

16) L133: what is the unit of measurement in the vertical axis of Figure 5A?

17) L165: “alter” should be corrected to “alters”

18) L166: correct “fastind” to “fasting”

19) L241: use “phenolic” instead of “phenol”

            In summary, the paper presents a compelling exploration of the effects of AOB on aging and lipid metabolism in C. elegans. With improvements in result presentation, figure resolution, and language clarity, this paper has the potential to make a significant contribution to the field.

Comments on the Quality of English Language

I strongly recommend a thorough English editing to improve the overall readability of the paper. The authors should consider professional editing services or having a native English speaker review the manuscript.

Author Response

 The paper investigates the potential positive effects of autumn olive berries (AOB) extract on aging delay by focusing on the improvement of lipid metabolism in middle-aged C. elegans, which have become obese due to a high glucose (GLU) diet. The study evaluates the antioxidant potential of freeze- and spray-dried AOB, with freeze-dried AOB demonstrating superior antioxidant capabilities, attributed to catechin as the main compound in the polyphenols of AOB. Subsequent in vivo experiments using freeze-dried AOB were conducted. The results overall imply that AOB may contribute to delaying aging by improving lipid metabolism in metabolic-impaired middle-aged worms.

            The paper covers an interesting topic with potential implications for understanding the role of natural compounds in lipid metabolism and aging. The manuscript is generally well-written, providing an overview of the research objectives, methods, and key findings. However, there are a few areas that need attention:

  1. Clarity of Results Presentation: The results need to be presented more clearly. Please check comments below.

The manuscript has been revised according to your comment. The revised part has been marked in red.

  1. Figure Resolution: The figures accompanying the paper appear somewhat blurry. It is recommended to enhance the resolution of all figures to ensure that the data is presented clearly and is easily interpretable.

We improved the resolution of the figures. However, if we put the whole worm's body in the box, the resolution may seem low. If necessary, we will zoom in on only a part of the worm's body.

  1. English Language Corrections: The manuscript could benefit from a thorough proofreading to correct grammatical errors, improve sentence structure, and ensure overall language clarity. This is essential for enhancing the readability of the paper.

The English editing of the manuscript was carried out and overall revised.

  1. Comments: Some comments also included that need to be considered:

The revised part of the author's opinion was marked in red.

1)    L3: “delays” should be corrected to “delay”

We modified it according to your comment.

2)    L3: use the full name for “C. elegans” in the title, in the abstract and when you mention it for the first time in the main text

We modified it according to your comment.

3)    L27: “contributes” should be corrected to “contribute”

We rewrote the sentence that includes 'contributes'.

4)    L54: “has effects” should be corrected to “have effects”

We rewrote the sentence that includes 'has effects'.

5)    L55: do you mean “to reduce obesity”?

We removed that sentence.

6)    L63: use “phenolic” instead of “phenol”, as you did in Table 1

We modified it according to your comment.

7)    L74: prefer “low” instead of “small”

We modified it according to your comment.

8)    L86-87: omit the duplicate sentence

Duplicate sentences were deleted.

9)    L89-95: paragraph should be rewritten; 1) which figure shows that survival decreased in the concentration range of 500-1000 μg/mL of AOB? 2) you mention that safe concentrations used were 62.5, 125 and 250 (Figure 2A), but in Figure 2A all concentrations are shown and appear to be similar

FAOB showed statistically no difference in survival rate in the range of 0–1000 μg/mL concentration. However, the survival rates decreased by 2 and 4% at 500 and 1000 μg/mL FAOB, respectively. Therefore, we excluded these two concentrations from the experimental concentration.

3) since you used FAOB, you would better write FAOB 250 in Figures 2B and 2C 4) the survival of nematodes was affected, but not compared to the control, so be specific adding “compared to the control”

We modified it according to your comment.

10) L102-108: 1) according to what you write under the Figure 3, Figure 3A, 3B and 3C inside the text should be changed to 3B, 3C and 3D, respectively 2) write TG in full name 3) TG Figure corresponds only to middle-aged worms according to what is written below, so be specific in lines 107-108

We modified it according to your comment.

11) L110-117: 1) what “-” and ”+” mean? it isn’t mentioned anywhere 2) middle-aged worms are shown in Figure 3C? please correct it in line 112 3) correct “C” to “D” and add “under normal and 2% GLU condition” 4) write TG in full name for the first time

We modified it according to your comment.

12) L119: write ROS in full name for the first time

We modified it according to your comment.

13) L122: please improve the quality of images presented in Figure 4

The image quality shown in Figure 4 has been improved.

14) L131: what do you mean “increased behavior”? please be more specific

We modified it to 'increased locomotion ability'.

15) L133: please improve the quality of images presented in Figure 5

The image quality shown in Figure 5 has been improved.

16) L133: what is the unit of measurement in the vertical axis of Figure 5A?

Number of body bending performed during 20 seconds.

17) L165: “alter” should be corrected to “alters”

We modified it according to your comment.

18) L166: correct “fastind” to “fasting”

We modified it according to your comment.

19) L241: use “phenolic” instead of “phenol”

We modified it according to your comment.

            In summary, the paper presents a compelling exploration of the effects of AOB on aging and lipid metabolism in C. elegans. With improvements in result presentation, figure resolution, and language clarity, this paper has the potential to make a significant contribution to the field.

Thanks for your specific comments. In the revised manuscript, all modified parts are marked in red.

Reviewer 2 Report

Comments and Suggestions for Authors

Author Response

Reviewer 2:

Several clarifications that should be made in the following manuscript:

Section Introduction

The full name of C. elegans should be provided in the title of the manuscript and in the first mention in the Abstract section.

We modified it as per your comment.

The Introduction section needs significant improvement with brief presentation of previous studies for the presence of bioactive compounds in autumn olive berries and their biological activities, particularly the effects on lipid metabolism.

We added more information about AOB effect on metabolic diseases. The additions were marked in red.

Pg. 2, Line 55: The following sentence is not clear and it should be revised: “In this experiment, we aim to improve obesity using this approach.

We removed that sentence.

Section Materials and Methods

The freeze- and spray-drying procedure for AOB samples should be presented.

We specifically presented the procedure for preparing FAOB and SAOB used in the experiment. The added part was marked in red.

Why the authors used water for extraction of phenolic compounds, since it is well known that alcohols (methanol, ethanol) or aqueous alcohols are commonly used solvent mixtures?

We aim to use AOB as tea. Therefore, we chose water extraction instead of alcohol or water-based alcohol extraction to proper our purpose of using AOB.

The most important things that are not presented in this study:

  • how the evaporated powder samples were processed before the assays?
  • what type of solvent was used for dissolving the dried extracts.

AOB samples were dissolved in DMSO to prepare a 15 mg/mL stock solution and were stored at -80ºC and then diluted fresh in distilled water to an appropriate concentration right before the experiment.

We added that information to the Materials and Methods section. The added parts are marked in red.

  • what type of dried extracts were used for phytochemical analyses (HPLC, TPC, DPPH) and antiobesity or antiaging activities?

The activities of FAOB and SAOB were evaluated in HPLC, TPC, and DPPH assays. Based on the results of HPLC, TPC, and DPPH, FAOB was determined to have more potential. Anti-obesity and anti-aging effects were verified with FAOB.

Several concerns in this study are related to the HPLC conditions:

  • The HPLC protocol is not appropriate for separation of many compounds. The comparison of the chromatograms on Figure 1 between methanol and the samples showed that only one clear peak is present in the samples. However, it cannot be state that this compound is catechin only on the basis of standard catechin.

We agreed with your comment and analyzed it again under different HPLC analysis conditions. However, even when analyzed again, there was one significant large peak of AOB, and this peak had the same retention time as catechin. By analyzing the DAD spectrum of the peak, it was confirmed that the large peak of AOB indicated catechin. It is considered that additional analysis through MS analysis is necessary for the unknown small peak, which has not been identified as what substance it is.

  • Why the authors used only catechin as a standard?
  • Identification of phenolic compounds cannot be performed without confirming the compounds by mass spectrometry.

A literature review confirmed that AOB contained phenolic substances such as catechin, gallic acid, caffeic acid, quercetin, and lutein, so we performed a qualitative analysis of HPLC on these substances. As a result, it was confirmed that only catechin was detected in the AOB samples we used.

Why the reaction mixture for TPC was read on 720 nm, since the blue color due to reduction of Folin-Ciocalteu should be read on 765 nm?

The wavelength used for TPC analysis is set differently for each study. Generally, the wavelength is set to 650-770 nm for TPC analysis. The sample AOB has a red color and affects color development during the reaction process with Folin-Ciocalteu reagent. Thus, it was measured at 765 nm to avoid this interference.

In the method for TPC, the authors state: “The results were expressed as gallic acid equivalents (GAE) per gram”, but in Table 1, the results are shown in mg GAE/100 g.

We corrected the unit for TPC in the Materials and Methods section.

Expression of the results for DPPH assay should also be presented.

We presented a representation of the results of the DPPH analysis in the Materials and Methods section.

Section Results

The TPC content in FAOB sample is 0.26 mg GAE/100 g, while only catechin content is 10.93 µg/g. It seems that TPC content in the sample is too small, even smaller than catechin. Any explanation for this inconsistency? 

We recalculated the TPC content and found some errors. In addition, catechin content has changed from previous versions due to improved conditions for analyzing catechins. The changes are marked in red.

The authors state that “The results show that catechin was the key compound of FAOB”. How the authors make a screening of the compounds and concluded that catechin is the major compound, when there is no detection of other compounds?

A literature review confirmed that AOB contained phenolic substances such as catechin, gallic acid, caffeic acid, quercetin, and lutein, so we performed a qualitative analysis of HPLC on these substances. As a result, it was confirmed that only catechin was detected in the AOB sample we used. In further research, it is necessary to perform more accurate profiling through MS analysis. Accordingly, further research was suggested in the discussion section.

Also, the statement “catechin peak appeared high in FAOB and small in SAOB“ could not be considered relevant, since the peak for catechin in SAOB sample did not correspond to those in FAOB sample and the standard.  

In the result of the second HPLC analysis, there was one significant large peak of AOB, and this peak had the same retention time as catechin. By analyzing the DAD spectrum of the peak, it was confirmed that the large peak of AOB indicated catechin.

From the Figure 2A, it can be seen that there is no differences in the survival rate upon different extract concentration. Since the authors state that: “However, worm survival decreased in the concentration range of 500-1000 μg/mL of AOB.”, it should be presented how much is the reduction of survival rate in percentage for those high extract concentrations compare to control. Are these reductions in survival rate significant?

FAOB showed statistically no difference in survival rate in the range of 0–1000 μg/mL concentration. However, the survival rates decreased by 2 and 4% at 500 and 1000 μg/mL FAOB, respectively. Therefore, we excluded these two concentrations from the experimental concentration.

The authors state that: “Also, FAOB did not affect the survival of nematodes even under oxidative and heat stress conditions (Figure 2B and 2C)“, but the results in these Figures are related to AOB sample. What is really used in this assay?

FAOB was used in all of our in vivo studies. All parts written AOB were modified to FAOB. The revised part was marked in red.

Figure 3: There is a mistake in the Figure 3 Caption indication for 3B,C and D. The explanation for Figure 3C in the caption is related to Figure 3D

I revised the part you pointed out.

Pg. 4, Line 119-120: The following sentence should be revised: “ROS is a metabolite that causes aging, and it has been reported in previous studies that a high-GLU diet causes ROS accumulation [15].”, since ROS is not a metabolite…

Thank you for your comment. However, the definition of ROS is a metabolite is considered correct. Several papers support our opinion.

  • Dong, L.; He, J.; Luo, L.; Wang, K. Targeting the interplay of autophagy and ROS for cancer therapy: An updated overview on phytochemicals. Pharmaceuticals 2023, 16 (1), 92.
  • Jamieson, D. Oxygen toxicity and reactive oxygen metabolites in mammals. Free Radical Biology and Medicine 1989, 7 (1), 87-108.

The statement “Statistical analysis was performed using one-way ANOVA followed by Tukey’s post hoc test.” in Figure 3-7 Captions should be deleted and included in the Statistical analysis subsection in Material and Methods.

As per your comment, in the caption of Figure 3-7, the phrase "statistical analysis was performed using Tukey's post hoc test after one-way analysis of variance" was deleted.

Section Discussion

The main hypothesis of the authors is that AOB samples containing catechin effectively prevent obesity and delay aging. Taking into account the literature data that these plant samples represent a rich source of various phenolics and other phytochemicals, it cannot be considered that catechin is main compound responsible for these effects. In addition, the content of catechin in the sample that was used for in vivo experiments is only 10 µg/g. Do you believe that the content of catechin in tested extract concentrations (62.5, 125, and 250 μg/mL) might be responsible for such great bioactivity. The main disadvantage of this study is inappropriate phytochemical analysis of tested sample for the experiments including extraction of phenolics from the plant material. Despite this, the authors did not discuss the potential contribution of catechin or other phenolics characteristic for this sample to the lipid metabolism or aging processes.   

Based on your advice, we have modified the catechin analysis condition and as a result, the catechin content is increased than the values in the previous version.

We also added to the Discussion section that the phenolic or catechin content of AOB may change depending on growing conditions and post-harvest quality control and processing, and that research is needed on the effects of these pretreatments on functionality. The revised part was marked in red.

Thanks for your specific comments. In the revised manuscript, all modified sentences are marked in red.

Reviewer 3 Report

Comments and Suggestions for Authors

Reviewer comments:

In this work, Kim et al. investigated the effect of autumn olive berries on lipid accumulation, menopausal symptoms (reduced behavior, decreased fertility, ROS accumulation) and lipid metabolism (lipogenesis and lipolysis) related genetic factors in nematodes fed a high-glucose diet by middle age. Kindly, find below my comments for your response.

Introduction

Line 47-48: Kindly, revise the sentence.

The authors should highlight other previous studies that have used the berries as an intervention against a chronic disease or its potential application in food.

The authors should kindly highlight the use of the worm as a model for the assessment of the outcomes carried out. What makes them an ideal model for investigating the authors’ outcomes of interests? How are they related to humans or other animal models?

The mechanism around the feeding of high GLU diets to the worms and its associated increased ROS and obesity should also be highlighted.

In the abstract, the authors mentioned that two different forms (freeze-dried and spray-dried) were investigated. However, one of them was used. The authors should create a paragraph and highlight the effect of processing i.e freeze drying and spray drying on the nutritional and bioactive composition of the berries.

Methods

How many replicates of the TPC, DPPH and catechin analysis were conducted on the berries? This can give an idea of the replicates the means and SD were calculated from.

In the profiling of the phenolics, what informed the use of only catechin as the standard but not others like epicatechin etc?

Line 238: how much of the catechin was dissolved?

The authors should indicate that the DPPH was expressed AA/gram in the method section.

Line 262: 4.6.2. Acute toxicity

Why was cholesterol added as a composition of the buffer? The authors should make reference to previous studies that have used similar protocol.

In the toxicity studies, the authors didn’t make any reference to an established protocol. Could they address that? Do the authors think the observation of worm survival following the application of varying concentrations of the extract scientifically valid? What if internally certain tissues were affected? Is there a biomarker for assessing that?

Line 299: 4.6.7. Determination of ROS level

Can the authors make reference to a previous study that used same? How appropriate is the use of the microscope approach? Isn’t there a biomarker that could have been targeted? For example oxidised glutathione for example in humans?

Statistical analysis

Was data normality checked?

Results

I see that the Triglyceride is attached to Fig. 3D. However, in the statement preceding the Figures, reference to Triglyceride is made to Figure 3C. The authors should kindly correct that. Also, in Figure 3A, even though reference to Fat accumulation is made in the statements preceding the Figure, the amount of fat accumulated cannot be seen from the Figure.

Section “2.5. AOB reduces ROS accumulation in middle-aged C. elegans”

How did the authors confirm that truly high GLU-diet fed to the worms resulted in ROS accumulation?

Figure 5: How did the authors use the T-test? I see four different treatments in that Figure for the various days.

Discussion

The authors should present it in the way the results have been presented. Thus, the discussion on the berry should be introduced in any of the earlier paragraphs.

Author Response

Reviewer 3:

Reviewer comments:

In this work, Kim et al. investigated the effect of autumn olive berries on lipid accumulation, menopausal symptoms (reduced behavior, decreased fertility, ROS accumulation) and lipid metabolism (lipogenesis and lipolysis) related genetic factors in nematodes fed a high-glucose diet by middle age. Kindly, find below my comments for your response.

Introduction

Line 47-48: Kindly, revise the sentence.

The authors should highlight other previous studies that have used the berries as an intervention against a chronic disease or its potential application in food.

We added to the introduction the historical background of AOB being used as food and the results of previous studies on the benefits they indicate on metabolic diseases. The additions were marked in red.

The authors should kindly highlight the use of the worm as a model for the assessment of the outcomes carried out. What makes them an ideal model for investigating the authors’ outcomes of interests? How are they related to humans or other animal models?

We added the reason for using nematodes as experimental models. In particular, we explained that their genetic mechanisms related to aging are similar to mammals, including humans. Thus, C. elegans provides insight into understanding metabolic diseases in humans.

The mechanism around the feeding of high GLU diets to the worms and its associated increased ROS and obesity should also be highlighted.

Per your advice, we emphasized in the introduction that a high-GLU diet causes metabolic disorder symptoms, including obesity. The additions were marked in red.

In the abstract, the authors mentioned that two different forms (freeze-dried and spray-dried) were investigated. However, one of them was used. The authors should create a paragraph and highlight the effect of processing i.e freeze drying and spray drying on the nutritional and bioactive composition of the berries.

Following your advice, we highlighted the excellence of FAOB in the abstract and added the reason for using FAOB in the subsequent in vivo tests. The added part was marked in red.

Methods

How many replicates of the TPC, DPPH and catechin analysis were conducted on the berries? This can give an idea of the replicates the means and SD were calculated from.

We have performed three independent experiments on TPC and DPPH, and include that information in figure caption.

In the profiling of the phenolics, what informed the use of only catechin as the standard but not others like epicatechin etc?

A literature review confirmed that AOB contained phenolic substances such as catechin, gallic acid, caffeic acid, quercetin, and lutein, so we performed a qualitative analysis of HPLC on these substances. As a result, it was confirmed that only catechin was detected in the AOB sample we used. In further research, it is necessary to perform more accurate profiling through MS analysis. Accordingly, further research was suggested in the discussion section.

Line 238: how much of the catechin was dissolved?             

We analyzed catechin in the concentration range of 5-50 ug/mL and used a catechin calibration curve to calculate the catechin content of AOB. This information was indicated in the 'Materials and Methods' section.

The authors should indicate that the DPPH was expressed AA/gram in the method section.

We indicated the information according to your comment.

Line 262: 4.6.2. Acute toxicity

Why was cholesterol added as a composition of the buffer? The authors should make reference to previous studies that have used similar protocol.

Cholesterol was utilized due to its essential role for worms. Similar approaches are employed in other literatures.

  • Worm book: Maintenance of elegans (http://www.wormbook.org/chapters/www_strainmaintain/strainmaintain.html)
  • Kurzchalia, T. V.; Ward, S. Why do worms need cholesterol? Nature cell biology 2003, 5 (8), 684-688.
  • Kim, Y.; Lee, S.-b.; Cho, M.; Choe, S.; Jang, M. Indian Almond (Terminalia catappa Linn.) Leaf Extract Extends Lifespan by Improving Lipid Metabolism and Antioxidant Activity Dependent on AMPK Signaling Pathway in Caenorhabditis elegans under High-Glucose-Diet Conditions. Antioxidants 2023, 13 (1), 14.

In the toxicity studies, the authors didn’t make any reference to an established protocol. Could they address that? Do the authors think the observation of worm survival following the application of varying concentrations of the extract scientifically valid? What if internally certain tissues were affected? Is there a biomarker for assessing that?

The OECD (https://one.oecd.org/document/env/jm/mono(2015)16/part7/en/pdf) provides applications for toxicity assessment of nanomaterials using C. elegans. The OECD indicates a 24-hour exposure test with a short-term toxicity assessment using worms. However, in our study, the extract treatment period may extend to 48 h depending on the experiment, so the toxicity test period was extended to 48 h to perform a toxicity assessment. The source about we referenced has been added to the Materials and Methods section.

Line 299: 4.6.7. Determination of ROS level

Can the authors make reference to a previous study that used same? How appropriate is the use of the microscope approach? Isn’t there a biomarker that could have been targeted? For example oxidised glutathione for example in humans?

There are many studies that measure the intracellular ROS content using microscopes. These studies support the appropriateness of our experimental methods.

Kim, Y.; Lee, S.-b.; Cho, M.; Choe, S.; Jang, M. Indian Almond (Terminalia catappa Linn.) Leaf Extract Extends Lifespan by Improving Lipid Metabolism and Antioxidant Activity Dependent on AMPK Signaling Pathway in Caenorhabditis elegans under High-Glucose-Diet Conditions. Antioxidants 2023, 13 (1), 14.

Statistical analysis

Was data normality checked?

The statistical analysis was conducted using student’s t-test (unpaired, two-tailed) and one-way analysis of variance (ANOVA) followed by Tukey’s multiple range test. Also, Kaplan-Meier method (OASIS application,https://sbi.postech.ac.kr/oasis/) was used on survival results.

Results

I see that the Triglyceride is attached to Fig. 3D. However, in the statement preceding the Figures, reference to Triglyceride is made to Figure 3C. The authors should kindly correct that. Also, in Figure 3A, even though reference to Fat accumulation is made in the statements preceding the Figure, the amount of fat accumulated cannot be seen from the Figure.

  • We corrected it fallowing your comment.
  • Oil Red O, a red dye utilized for fat staining, demonstrates in Figure 3A that the red color lightens with AOB ingestion. This means AOB inhibited fat accumulation.

Section “2.5. AOB reduces ROS accumulation in middle-aged C. elegans”

How did the authors confirm that truly high GLU-diet fed to the worms resulted in ROS accumulation?

As shown in figure4, there is a difference in the intensity of the green signal between GLU (-) and (+). This means that GLU affects ROS expression.

In addition, in our previous study, it was confirmed that GLU induces significant oxidative damage, such as juglone.

  • Kim, Y.; Lee, S.-b.; Cho, M.; Choe, S.; Jang, M. Indian Almond (Terminalia catappa Linn.) Leaf Extract Extends Lifespan by Improving Lipid Metabolism and Antioxidant Activity Dependent on AMPK Signaling Pathway in Caenorhabditis elegans under High-Glucose-Diet Conditions. Antioxidants 2023, 13 (1), 14.

Figure 5: How did the authors use the T-test? I see four different treatments in that Figure for the various days.

We first performed ANOVA analysis on the total body bending. However, it was confirmed that there was no statistical significance, even though an increase in the sample treatment group was observed compared to the control.

Therefore, a t-test of the treatment group and the control for each concentration was performed after that. As a result, a difference was confirmed between the control and the sample for each concentration.

The display format has been modified to reduce confusion in displaying statistical differences above the bars.

Discussion

The authors should present it in the way the results have been presented. Thus, the discussion on the berry should be introduced in any of the earlier paragraphs.

We restated our opinion (discussion) in the order in which the results were presented.

And, thanks for your specific comments. In the revised manuscript, all modified sentences are marked in red.

Round 2

Reviewer 2 Report

Comments and Suggestions for Authors

The authors satisfactorily responded to all suggestions. 

Reviewer 3 Report

Comments and Suggestions for Authors

Thank you for making time to undertake the revisions which has led towards an improvement in the quality and intent of the work. I am happy with the revisions. Well done to the team!